Synchrotron µ-XRF mapping analysis of trace elements in in-situ cultured Japanese red coral, Corallium japonicum

Iwasaki Nozomu 1 iwasakin@ris.ac.jp
Hasegawa Hiroshi 2
http://orcid.org/0000-0001-8004-895X Tamenori Yusuke 3
Kikunaga Mutsuro 4
Yoshimura Toshihiro 5
Sawai Hikaru 6
1 Faculty of Geo-Environmental Science, Rissho University , Kumagaya, Saitama , Japan
2 Institute of Science and Engineering, Kanazawa University , Kanazawa, Ishikawa , Japan
3 Japan Synchrotron Radiation Research Institute , Sayo, Hyogo , Japan
4 Fukada Salvage & Marine Works Co., Ltd. , Tokyo , Japan
5 Research Institute for Marine Resources Utilization, Japan Agency for Marine-Earth Science and Technology , Yokosuka, Kanagawa , Japan
6 Ibaraki College, National Institute of Technology , Hitachinaka, Ibaraki , Japan
Reimer James
Electronic publication date: 2022 Aug 23
Publication date: 2022
Volume: 10
Electronic Location ID: e13931
Received 2022 Mar 10; Accepted 2022 Jul 31
Copyright: © 2022 Iwasaki et al.
Copyright year: 2022
Copyright holder: Iwasaki et al.
License: This is an open access article distributed under the terms of the Creative Commons Attribution License, which permits unrestricted use, distribution, reproduction and adaptation in any medium and for any purpose provided that it is properly attributed. For attribution, the original author(s), title, publication source (PeerJ) and either DOI or URL of the article must be cited.
License URL: https://creativecommons.org/licenses/by/4.0/

Keywords: Synchrotron radiation spectroscopy, µ-XRF/XAS, Japanese red coral, Corallium, Growth rate, Biogenetic carbonate

Funding: Research and Development Projects for Application in Promoting New Policy of Agriculture, Forestry and Fisheries 22032 Ministry of Agriculture and Fisheries, Japan The work was supported by the Research and Development Projects for Application in Promoting New Policy of Agriculture, Forestry and Fisheries (22032), Ministry of Agriculture and Fisheries, Japan. The funders had no role in study design, data collection and analysis, decision to publish, or preparation of the manuscript.

==============================
Precious corals belong to the family Coralliidae (Cnidaria, Octocorallia), and their axis, which consists of high magnesian calcium carbonate, has long been used in jewelry. With its low growth rate and long lifespan, precious coral is a representative taxon of the vulnerable marine ecosystem. Due to years of overfishing, coral fishery has become a controversial issue. To estimate the growth rate and clarify the uptake process of trace elements in relation to the growth of the carbonate axis, Japanese red coral (Corallium japonicum) was cultured at a depth of 135 m off Takeshima Island, Kagoshima, Japan for 98 months and analyzed by microscopic X-ray fluorescence/soft X-ray photoabsorption (µ-XRF/XAS) speciation mapping. The growth rate was estimated to be 0.37 mm/year in diameter, and 10–11 growth rings were observed in a cross section of the axis. This estimated growth-rate value is the first ever to be obtained from the in-situ culture of Japanese precious coral. The fluctuation in water temperature near the in-situ-culture site was recorded for part of the culture period and then compared with the changes in the growth ring and the distribution of trace elements in a cross section of the coral axis during the same period. When the water temperature was increasing, the growth ring was light in color, sulfur and phosphorus concentrations were low, and magnesium was high. Conversely, a dark band in the growth ring, high sulfur and phosphorus, and low magnesium concentrations were observed when the water temperature was decreasing. In a cross section of the coral axis, the distribution of sulfur and magnesium from the center to the surface corresponded, respectively, to dark and light bands in the annual growth ring. Sulfur concentration was high in the dark band and low in the light band, while magnesium was negatively correlated with sulfur.

Introduction

Precious corals belong to the family Coralliidae (Cnidaria, Octocorallia), and their axis, which consists of high-magnesian calcium carbonate, has long been used in jewelry. Japanese waters are one of the world’s major coral fishing grounds, and corals of high quality are harvested there. Distributed at depths of 80 to 300 m, Japanese red coral (Corallium japonicum), pink coral (Pleurocorallium elatius) and white coral (P. konojoi) have been harvested from Japanese waters for 150 years. Fishing pressure has increased in recent years on the back of soaring precious-coral prices, and concerns have been raised regarding the effect of fishing pressure on the depletion of precious coral resources (Iwasaki, 2019). The Japanese Ministry of the Environment has therefore designated the three above-mentioned species as “Near Threatened”. The fishing and conservation of coral resources have become controversial international issues. In August 2019, the 18th Conference of the Parties of the Convention on International Trade in Endangered Species of Wild Fauna and Flora (CITES) adopted a decision to compile recommendations on the conservation and sustainable use of precious corals (Decision 17.192, Rev. CoP18). The Food and Agriculture Organization of the United Nations (FAO) has consequently reviewed the fishing and global trade of precious corals and summarized the issues regarding its sustainable use (Cannas et al., 2019).

For the sustainable use of precious corals, management according to a population dynamics model is indispensable, and this requires estimates of biological parameters such as recruitment, growth and mortality rates. The purpose of this study is to clarify the formation process of the carbonate axis and to estimate its growth rate. Clarification of the environmental factors affecting axis growth not only contributes to the sustainable use of resources but also provides valuable information regarding the effects of environmental changes, particularly ocean warming and acidification, on carbonate-axis formation in marine organisms.

The axis of precious corals is composed of calcium carbonate crystallized as calcite and contains rare elements such as magnesium, strontium, barium, iodine, molybdenum, manganese, zinc, cadmium, and bromine (Hasegawa et al., 2010; Iwasaki et al., 2014). In addition, close to 2 wt% (ranging from 1.8 to 4.6 wt%) of the axis is made up of organic matter, such as proteins, lipids and sugars (Allemand et al., 1994; Vielzeuf et al., 2018). The concentrations of trace elements taken from the seawater into the axis and the organic matter synthesized by the precious coral fluctuate periodically with its growth. According to previous studies on the growth mechanism of red coral (C. rubrum), the nascent axis and its tips are formed by aggregations of sclerites secreted by scleroblasts in the coenenchyme, which are then surrounded by deposits of calcium carbonate secreted by the axis epithelium (Allemand, 1993; Grillo, Goldberg & Allemand, 1993; Allemand & Bénazet-Tambutté, 1996; Perrin et al., 2015; Urushihara, Hasegawa & Iwasaki, 2016). The axis grows lengthwise while expanding radially outward from the center. Concentric growth rings can therefore be observed in a thin cross section of the axis using a petrographic method of direct microscopic observation. These rings have been used to estimate the growth rate of the axis. Marschal et al. (2004) developed a new technique to estimate the growth rate using the chemical substance toluidine blue. When stained with toluidine blue, each growth ring in the axis of C. rubrum has been seen to consist of two bands: a thin, dark-colored band and a thick, light-colored band. The former contains more organic matrix than the latter and is formed during the late fall and winter months, which correspond to the slow-growth season. The two bands are formed over the course of a year (Marschal et al., 2004). In our study, we demonstrated that similar growth rings can be observed with a petrographic method using a high-resolution digital microscope even without staining the axis (Luan et al., 2013). Furthermore, we confirmed that the distribution pattern of sulfur, phosphorus and magnesium in the axial skeleton of C. japonicum corresponds to dark/light bands along the annual growth rings. Inorganic sulfate and phosphorus concentrations are lower and magnesium concentration is higher in the thin, dark bands, which are richer in organic matrix than the thick, light bands (Luan et al., 2014; Tamenori et al., 2014). The concentrations of trace elements in the axis reflect their concentrations in the habitat seawater, which vary depending on physical and chemical environmental factors, such as water temperature. It is known that the concentrations of barium and lead in the axis reflect their concentration in the habitat seawater, and the concentration of magnesium in the axis changes depending on the water temperature (Hasegawa et al., 2010; Hasegawa et al., 2012; Vielzeuf et al., 2018; Weinbauer, Brandstätter & Velimirov, 2000; Vielzeuf et al., 2021).

We investigated the distribution of trace elements in the axis of in-situ cultured C. japonicum using X-ray spectroscopic techniques combined with micro-focused soft X-ray radiation, i.e., microscopic X-ray fluorescence/soft X-ray photoabsorption (µ-XRF/XAS) speciation mapping. In addition to discussing the relationship between the uptake of trace elements into the coral axis and its growth, we compared two-dimensional images of the distribution of trace elements in the coral-axis cross section with the habitat water temperature.

Materials and Methods

In-situ culture of Japanese red coral

The samples analyzed in this study were Japanese red coral (C. japonicum), which had been cultured in situ for 98 months. Four colonies of C. japonicum were collected from the waters north of Iwojima, Kagoshima, southern Japan on 26 March 2005. They were bonded to an iron table on board the vessel the same day and placed on the seafloor at a depth of 135 m off Takeshima, Kagoshima (30°50.40N, 130°23.97E). The water temperature, current velocity and direction were 16 °C, 0.3–05 kt and 90–110°, respectively. The corals were observed in situ on 25 July 2006, 18 June 2007, 24 September 2007 and 29 April 2011 and collected on 21 May 2013. The in-situ culture was conducted by the manned submersible Hakuyo, owned by Fukada Salvage & Marine Works Co., Ltd. Permission to collect precious corals was obtained from the Governor of Kagoshima Prefecture, Japan (#KAKOSANGO 19-1).

The water temperature was measured near the table at 135 m depth (30°50.40N, 130°24.12E) using an Onset HOBO TidbiT v2 temperature data logger. The water temperature was recorded every 15 min from 4 November 2012 to 28 January 2014 and 10-day moving averages were calculated. The period of water-temperature measurement overlapped with the in-situ culture for 6.5 months from 4 November 2012 to 21 May 2013. To compare the fluctuation in water temperature with changes in the growth rings and trace-element concentrations, sections corresponding to the axis growth during those 6.5 months were cut out from the element-concentration mapping images. The sections were removed from areas where there were no cracks in the cross section. In the images of the growth rings and the trace-element-concentration mapping, the direction of growth is from left to right, so the images were flipped horizontally to compare them with the fluctuation in water temperature.

Estimation of the growth rate of in-situ Japanese red coral

To estimate the growth rate, a branch where distinct growth was confirmed during the in-situ culture was used, i.e., the branch had a clear starting point from which it had grown until the time it was collected. The branch used for this study was an area of new growth on the right-hand side of colony B, one of the four cultured colonies (Fig. 1, B-1, I). Branch I, which was small and knob-like, measuring 1.95 mm from the trunk at the start of the study (Fig. 1A, B-1, I), grew and branched into two during the in-situ culture (Fig. 1B, V and VI). The growth rate of branch I was estimated by comparing its size before and after the in-situ culture. To estimate the linear growth rate, the axis was measured lengthwise from the original tip of branch I to the tips of branches V and VI. The diameter of cross-section slab “b”, as described later, was measured to estimate the radial growth rate (Fig. 1B).

Figure 1 Japanese red coral (Corallium japonicum) colony B cultured on the seafloor at 135 m depth off Takeshima Island, Kagoshima, Japan from 26 March 2005 to 21 May 2013.

(A) Before the in-situ culture experiment. (B) After the in-situ culture experiment; dried coenenchyme remains on the axis surface (the orange-colored area). The small branch I of B-1 grew to the axis tips V and VI during the in-situ culture experiment, while branches II–IV were lost.

Furthermore, branches II, III and IV were lost during the in-situ culture experiment (Fig. 2). When the coral was collected on 21 May 2013, it was noted that coenenchyme had grown over the exposed areas on the surface of the axis where the branches had broken off, confirming that the rest of the colony was still alive and growing.

Figure 2 Observations of the Japanese red coral (Corallium japonicum) colony B on the sea floor at 135 m depth off Takeshima Island, Kagoshima, Japan from 26 March 2005 to 21 May 2013 using the manned submersible Hakuyo.

(A) 26 March 2005, the first day of the in-situ culture experiment. (B) 25 July 2006. (C) 18 June 2007 (mirror-reversed image). (D) 24 September 2007. E: 29 April 2011. The arrows indicate the branches that were lost during the experiment.

Sample preparation for µ-XRF/XAS mapping analysis

Slabs for the cross-section and longitudinal-section samples were cut from the new growth on branch I (Fig. 1). The slab for the cross-section sample was cut transversely from part “b”, which corresponds to the tip of branch I before the in-situ culture (Fig. 1B, b; sample number B-1-b). The diameter and thickness of the cross-section sample were 3.22 and 5.0 mm, respectively. For the longitudinal section, the slab was cut vertically along part “c”, which extends out from “b” in B-1 (Fig. 1B, sample number B-1-c-1). The diameter, length and thickness of the longitudinal-section sample were 3.16, 7.40 and 5.0 mm, respectively. The slabs were embedded in an unsaturated polyester resin and the cut surfaces were polished using 3MTM Lapping Film Sheet (#8000, alumina).

Dried coenenchyme, containing numerous sclerites, remained on the outer circumference of the cross-section slab and both sides of the longitudinal-section slab. Patches of sclerite aggregates were also distributed in a pit in the longitudinal-section slab where it cut through a cavity in the axis (Fig. 3).

Figure 3 Longitudinal section of the Japanese red coral (Corallium japonicum) cultured on the seafloor at 135 m depth off Takeshima Island, Kagoshima, Japan from 26 March 2005 to 21 May 2013.

(A) Longitudinal section of branch I of B-1-c (see Fig. 1B, c). The large white box indicates the area in which µ-XRF/XAS mapping analysis was performed (see Fig. 9). (B) Sclerites (s) and cavities of polyps (c) in the coenenchyme. (C) Patches of sclerite aggregates in a pit (p).

After µ-XRF mapping analysis, the cut surface of the cross-section sample (B-1-b) was polished further and the growth rings were observed using the high dynamic range of a VHX-1000 digital microscope (Keyence Corporation, Osaka, Japan) (Luan et al., 2013).

µ-XRF/XAS mapping analysis

Synchrotron radiation at SPring-8, Japan was used for the X-ray analysis of the distribution of trace elements in the coral axis. Microscopic X-ray fluorescence/soft X-ray photoabsorption (µ-XRF/XAS) speciation mapping was carried out at the B-branch of the soft X-ray photochemistry beamline (BL27SU). The µ-XRF/XAS measurements were taken at excitation photon energy of 2510.0 eV. The X-ray beam diameter was focused to a 10-µm spot on the sample point. The measurements followed the method described in Tamenori et al. (2014). The cross-section sample was irradiated at intervals of 10 μm length and 12 μm width within a range of 1.00 mm maximum length and 2.00 mm width. The longitudinal-section sample was irradiated at 40 µm intervals within a range of 3.16 mm length and 7.40 mm width. This study was conducted with the approval of the SPring-8 Proposal Review Committee (Proposals 2013B1301, 2014A1410 and 2015B1278).

Results

µ-XRF/XAS mapping analysis on the cross-section sample

Figure 4 shows two-dimensional µ-XRF/XAS mapping images of sulfur, magnesium and phosphorus concentrations in the cross section of the coral axis. Alternating layers of high and low sulfur and magnesium concentrations appear repeatedly in a concentric pattern. Their distributional patterns correspond to the dark and light bands in the annual growth rings in the microscopic images (Fig. 4D; Fig. 5). Sulfur concentration is high in the dark bands and low in the light bands, while magnesium is negatively correlated with sulfur, that is, high in the light bands and low in the dark bands. While less pronounced than that of sulfur or magnesium, the distributional pattern of phosphorus resembles that of sulfur, and is slightly higher in the dark bands. This study has clarified that the concentration of each of the three trace elements fluctuates periodically with the radial growth of the calcium carbonate axis from the center outwards.

Figure 4 µ-XRF/XAS mapping analysis of trace elements and microscope image in the cross section of the Corallium japonicum axis.

(A–C) Elemental mapping of magnesium, sulfur and phosphorus, respectively. (D) Image of the cross section observed under a high-resolution digital microscope; the inverted Y-shaped image is the center of the axis. The color bar represents the signal intensity, with blue and red corresponding to lower and higher concentrations, respectively. The arrows indicate the area (the white boxes) compared with the fluctuation in water temperature in Fig.10.

Figure 5 Growth rings in the cross section of the Corallium japonicum axis observed under a high-resolution digital microscope.

The dashed lines indicate dark bands in the growth rings. The solid line indicates the transect for distributions of magnesium, sulfur and phosphorus (see Figs. 6 and 7).

In the center of the axis, the concentrations of magnesium and sulfur are high and low, respectively (Fig. 4, inverted Y-shaped images). In the coenenchyme on the axis surface, where numerous sclerites are distributed, the concentration of each of the three trace elements is high in some areas.

Fluctuations of magnesium, sulfur and phosphorus along a transect which is drawn from the center to the surface on a cross section without cracks are shown in Fig. 6. There is high magnesium and low sulfur and phosphorus in the center of the axis (−0.2–0.2 mm) and around 0.7–0.8 mm from the center. The ratio of magnesium to sulfur shows a clear negative correlation (Fig. 7). The areas where the Mg/S ratio is low, that is low magnesium and high sulfur, roughly correspond to the center of the axis and the dark bands in the growth rings. The relationship between the fluorescence signals of magnesium and sulfur along the transect also shows a negative correlation (Fig. 8).

Figure 6 Fluctuations in magnesium, sulfur and phosphorus along the transect in the cross section of the Corallium japonicum axis (see Fig. 5).

The direction of the x-axis is opposite to the transect in Fig. 5.

Figure 7 Fluctuation in the magnesium/sulfur ratio (fluorescence signals) along the transect in the cross section of the Corallium japonicum axis.

The direction of the x-axis is opposite to the transect in Fig. 5. A–F indicate dark bands in the growth rings (see Fig. 5).

Figure 8 Relationship between the fluorescence signals of magnesium and sulfur at the transect in the cross section of the Corallium japonicum axis (see Fig. 5).

µ-XRF/XAS mapping analysis on the longitudinal-section sample

In the longitudinal section, µ-XRF/XAS mapping images of sulfur, magnesium, phosphorus and oxygen concentrations were obtained (Fig. 9). The fluctuations in sulfur, magnesium and phosphorus concentrations appear as vertical, striped patterns lengthwise along the axis. Sulfur and magnesium in particular show noticeable variation and inverse correlation. On the other hand, oxygen concentration is almost uniform and shows no significant variation.

Figure 9 µ-XRF/XAS mapping analysis of trace elements in the longitudinal section of the Corallium japonicum axis (see Fig. 3).

The color bar represents the signal intensity, with blue and red corresponding to lower and higher concentrations, respectively.

In the coenenchyme on the left-hand side of the axis surface, sulfur, magnesium, phosphorus and oxygen concentrations are each high in some areas. Oxygen concentration is also high in the aggregated sclerites on the inner surface of the cavity.

In-situ growth rate

From branch I of B-1, which was small and knob-like at the beginning of the in-situ culture, to the tips of branches V and VI (Fig. 1), the axis measured 12.27 and 25.82 mm, respectively. The linear growth rates were therefore estimated to be 1.50 and 3.16 mm/year, respectively. Meanwhile, the diameter of cross-section slab “b” was 3.03 mm. This means the tip of branch I grew to an axis diameter of 3.03 mm during the in-situ-culture period of 98 months, an estimated radial growth rate of 0.37 mm/year. Furthermore, 10–11 growth rings were observed in cross-section slab “b” using the high-resolution digital microscope (Fig. 5).

The water temperature fluctuated from 13.1 °C to 22.0 °C with an average of 16.9 ± 1.1 °C during the 6.5-month period of the in-situ culture experiment. The 6.5-month period roughly corresponded to one fluctuation cycle of the water temperature in the 10-day moving average (Fig. 10B). To compare the images of the growth ring and the trace-element concentrations with the fluctuation in water temperature during the in-situ culture, the images were aligned with the period the water temperature was recorded (Fig. 10A). That is, the surface of the axis was placed to align with May 2013, and 0.1 mm in from the surface toward the center of the axis (0.1 mm being the estimated growth over 6.5 months) was placed at November 2012 (Fig. 10). The result confirmed that the formation of dark/light bands and the fluctuations in magnesium, sulfur and phosphorus concentrations roughly corresponded to the fluctuation in water temperature during the in-situ culture. From November 2012 to February 2013, when the water temperature was relatively high (average temperature 17.1 °C), the growth ring was light in color, magnesium concentration was high, and sulfur and phosphorus were low. On the other hand, from February to May 2013, when the water temperature was relatively low (average temperature 16.5 °C), the growth ring was dark, magnesium concentration was low, and sulfur and phosphorus were high.

Figure 10 Trace element concentrations in a growth ring in the cross section of the Corallium japonicum axis and their relationship with water temperature.

(A) Cross section of the axis observed under a high-resolution digital microscope; µ-XRF/XAS mapping analysis of magnesium, sulfur and phosphorus (mirror-reversed images of the area indicated by arrows in Fig. 4). (B) Fluctuation in water temperature and its 10-day moving average on the seafloor at 135 m depth off Takeshima Island, Kagoshima, Japan, from 4 November 2012 to 28 January 2014. The red line indicates the 10-day moving average during the in-situ-culture experiment, 4 November 2012 to 20 May 2013.

Discussion

Estimation of growth rate

This study estimated the growth rate of C. japonicum cultured in situ for 98 months and clarified fluctuations in magnesium, sulfur and phosphorus concentrations in the axis related to its growth. Although the growth rate of Japanese precious coral has been estimated in the past using harvested or collected coral, this study is the first to use in-situ cultured coral. This study found that the radial growth rate (diameter) of the in-situ cultured C. japonicum was 0.37 mm/year. Previous studies have estimated the radial growth rate of Japanese precious corals as follows: 0.22 mm/year for C. japonicum using the synchrotron radiation-infrared method and 0.20–0.27 mm/year using the high-resolution digital microscope method (Luan et al., 2013, Iwasaki et al., 2014); 0.44 mm/year for P. konojoi using the high-resolution digital microscope method (Luan et al., 2013); and 0.30 mm/year for P. elatius using the high-resolution digital microscope and 210 Pb methods (Hasegawa & Yamada, 2010; Luan et al., 2013). In addition, the growth rate of C. rubrum was estimated at 0.35 mm/year using an in-situ experiment and calcein dyeing method (Marschal et al., 2004). Thus, the growth rate obtained in this study was within the range of the estimated growth rate of precious corals in previous studies. For C. japonicum, however, the growth rate estimated in this study was slightly higher than the values in previous studies. This is thought to be related to the comparatively young age (98 months) of the sample used in this study. It has been reported that the growth rate tends to be high in the early stage of axis formation and decreases as the axis grows (Priori et al., 2013; Vielzeuf et al., 2018). In the Mediterranean red coral C. rubrum, the growth of the center of the axis (medullar zone), where axial growth begins, is fast at a linear growth rate of ~2 mm/year. On the other hand, the growth of the peripheral part (annular zone) is slow at a radial growth rate of ~0.2 mm/year (Perrin et al., 2015). It is conceivable that the growth rate in this study was overestimated because a relatively large proportion of the cross section used in the analysis was occupied by the central portion, which was in the early stage of formation and had a high growth rate (Fig. 4D, inverted Y-shaped image).

Not only does the growth rate of C. rubrum differ between the medullar and annular zones, the growth rate of the dark and light bands in the annual growth ring is also different (Marschal et al., 2004). Vielzeuf et al. (2018) reported that concentrations of magnesium, natrium, strontium, lithium and uranium in a cross section of P. konojoi were high in the fast-growing central part of the axis and decreased toward the surface. They interpreted this as an indication that the growth rate decreased with enlargement of the axis. Suzuki, Inoue & Yokoyama (2010) reported that the Mg/Ca and Sr/Ca ratios in a cross section of P. konojoi collected from off Cape Muroto, southern Japan showed a decreasing tendency from the axis center to the surface with no periodic variation. Assuming a water-temperature dependency (Weinbauer, Brandstätter & Velimirov, 2000), this decrease in the Mg/Ca ratio corresponds to a cooling of about 1.7–2.5 °C. However, an increase in the oxygen-isotope ratio from the center to the surface of the axis in the same sample indicated a cooling of about 8 °C. Considering the coral’s deep-sea habitat, it has been pointed out that, rather than fluctuations in water temperature, this variation in the oxygen-isotype ratio may be due to a kinetic-isotope effect related to a variation in the coral’s growth rate (Suzuki, Inoue & Yokoyama, 2010). Thus, the growth rate, particularly a decrease in the growth rate, has been named as the possible reason for this phenomenon. From the above, it is thought that annual cycles of two different growth rates are repeated while the overall growth rate decreases with age. Therefore, it is conceivable that the growth rate of old colonies is overestimated while that of young colonies is underestimated. This assessment of growth rate is important in managing precious coral resources. For accurate estimation, it is effective to grasp the change in growth rate from the center of the axis to the periphery using the 210 Pb method (Hasegawa & Yamada, 2010).

Periodic fluctuation of rare element concentrations in a cross section of the axis

This study observed concentric growth rings from the center to the surface of the axis in a cross section of C. japonicum using a high-resolution digital microscope. Significant periodic variations corresponding to the growth rings were observed in the concentrations of sulfur and magnesium. As found in previous studies, sulfur concentration was high in the thin, dark bands of the growth rings and low in the thick, light bands, and magnesium was negatively correlated with sulfur (Luan et al., 2014; Tamenori et al., 2014). In addition, this study revealed that distinct fluctuations in sulfur and magnesium concentrations can also be seen in the form of vertically striped patterns in a longitudinal section of the axis. As a precious coral grows, the axis elongates while expanding radially from the center to surface, so the growth rings appear as vertical stripes in the longitudinal section and concentric rings in the cross section. As in the cross section, the vertically striped pattern formed by fluctuations in trace-element concentrations in the longitudinal section are thought to occur with the growth of the axis. This study clearly showed the periodic variation in trace elements in the growing axis by analyzing an entire cross section of the axis from the center to the surface as well as a longitudinal section. Vielzeuf et al. (2018) reported the vertically striped pattern formed by fluctuations in Mg, S and Na concentrations in a longitudinal section of white coral, P. konojoi. The present study has further demonstrated the periodical fluctuations of trace elements such as Mg and S with growth by analyzing both cross and longitudinal sections of the axis of C. japonicum.

Concerning the relationship between trace elements in the axis of C. rubrum and water temperature, Weinbauer, Brandstätter & Velimirov (2000) reported that the Mg/Ca ratio fluctuates periodically along the growth direction in a cross section of the axis. Concerning the Mg concentration in the coral axis, they reported that water temperature is positively correlated with magnesium concentration since higher water temperatures promote the incorporation of magnesium into the coral axis. They arrived at this conclusion by examining the Mg concentration and the habitat water temperature of C. rubrum collected from different depths. Moreover, Yoshimura et al. (2011) demonstrated a positive correlation between Mg/Ca and water temperature for precious corals collected from Japanese and Midway waters and the Mediterranean Sea.

It has been reported that Mg in the axis of C. rubrum fluctuates periodically with growth and is related to fluctuations in water temperature (Chaabane et al., 2019; Weinbauer, Brandstätter & Velimirov, 2000). And the relationship between the fluctuation in Mg concentration and annual and seasonal fluctuations in water temperature has been analyzed over a long period (Vielzeuf et al., 2013). However, although the periodical fluctuation in Mg concentration in the axis of Japanese precious corals with growth has been clarified, no previous studies have explored the relationship between Mg concentration and fluctuations in habitat temperature. In this study, we were able to compare the fluctuation in water temperature near the in-situ-culture site with the magnesium, sulfur and phosphorus concentrations in the growing part of the axis for 6.5 months of the 98-month in-situ-culture period. This study showed that the growth ring was light in color, sulfur and phosphorus concentrations were low, and magnesium was high during the high-water-temperature period. Conversely, the growth ring was dark, sulfur and phosphorus concentrations were high, and magnesium was low during the low-water-temperature period. Concerning the relationship between magnesium concentration and water temperature, this study confirmed that they are positively correlated.

As mentioned above, this study showed that there is a positive correlation between the Mg concentration in C. japonicum and the chronological change in water temperature in its habitat. However, Vielzeuf et al. (2013) showed that the Mg concentration in C. rubrum did not reflect a 1 °C rise in water temperature over 30 years. They suggested that the Mg concentration was not determined simply by water temperature but was influenced by the growth rate of the axis and growth interruption. With regard to magnesium in particular, it has been suggested that Mg uptake in C. rubrum is regulated by kinetic effects such as growth rate because the Mg content is abundant in the fast-growing bright band in the peripheral part (annular zone) of the axis and in the fast-growing center of the axis (medullar zone) (Vielzeuf et al., 2018; Chaabane et al., 2019).

Concerning the formation of dark and light bands in the peripheral part of the axis, Marschal et al. (2004) reported that the growth rings of C. rubrum consist of two bands, one dark and thin and the other light and thick; a new growth ring is formed annually, and the dark, thin band is formed during the late-autumn and winter months. They therefore suggested that the formation of the growth rings may be related to water temperature. In the present study, the growth ring was seen to be light when the water temperature was high and dark when the water temperature was low. However, five fluctuation cycles of water temperature were recorded from November 2012 to January 2014, demonstrating that the water temperature was not necessarily higher in summer and lower in winter (Fig. 10). Based on these findings, if the formation of dark and light bands is due to water temperature, more than one pair of bands should be formed each year. To clarify more precisely the effect of water temperature on dark/light growth-band formation and trace elements, further research—such as water-control experiments using cultured corals and determination of water temperatures by analyzing oxygen-isotope ratios in the dark/light bands—is required. Mass mortality events of C. rubrum are known to have occurred in the Mediterranean due to heat waves (Garrabou et al., 2001; Garrabou et al., 2009), but the effects of slow rises in temperature have yet to be predicted. To clarify how rising seawater temperatures due to global warming will affect the growth and axis-formation of precious corals, further research is needed on the role water temperature plays in growth-band formation and the concentrations of trace elements in the axis.

Negative correlation between Mg and S concentrations

This study showed that the concentrations of Mg and S in the axis of C. japonicum are negatively correlated. This relationship was reported in C. japonicum, C. rubrum and P. elatius by quantitative analyses using the µ-XRF and electron microprobe (EMP) (Vielzeuf et al., 2013; Luan et al., 2014; Tamenori et al., 2014). Regarding the sulfur detected in this study, the µ-XRF/XAS analysis revealed that this is mainly in the form of gypsum-like inorganic sulfate ions substituting for carbonate ions in the axis, and previous studies have indicated that inorganic sulfate is negatively correlated with Mg (Luan et al., 2014; Tamenori et al., 2014; Perrin et al., 2017). The sulfur detected in this study is thought to be inorganic because the analysis was performed under the same experimental conditions as previous papers reporting the detection of inorganic sulfur (Luan et al., 2014; Tamenori et al., 2014).

Yoshimura et al. (2015) suggested that Mg is incorporated into CaCO3 by ion substitution for calcium. The concentration of Mg in the axis is affected by the water temperature and the growth rate as described above. On the other hand, inorganic S is considered to be incorporated into the calcite skeleton as sulfate instead of carbonate ions (Tamenori et al., 2014). Regarding the negative correlation between Mg and S concentrations, Luan et al. (2014) point out that annual fluctuations in carbonate ion concentration have a negative effect on the uptake of sulfur into calcium carbonate. While this and previous studies have clarified this negative correlation as a phenomenon, further research is required to determine the contributing factors and the presence or absence of chemical antagonism.

Trace element concentrations in the center of axis and sclerites

In this study, the center of the axis (Fig. 4D, inverted Y-shaped image) was clearly distinguishable from the peripheral part, and the concentrations of magnesium and sulfur in the center were high and low, respectively. This is consistent with the results of previous studies (Perrin et al., 2015; Vielzeuf et al., 2018). Concerning the initial formation of the axis, previous studies reported that the cells and/or tissues involved in the formation of sclerites and axes were different (Allemand & Bénazet-Tambutté, 1996; Le Goff et al., 2017), and that the center of the axis was formed by aggregated sclerites (Allemand, 1993; Grillo, Goldberg & Allemand, 1993; Allemand & Bénazet-Tambutté, 1996; Perrin et al., 2015; Urushihara, Hasegawa & Iwasaki, 2016). Additionally, Perrin et al. (2015) indicated that the center of C. rubrum consisted of sclerites and sclerite aggregates embedded within calcite cement, while the annular parts were made up of fine concentric layers of calcite crystallites. If the center of the axis originates from sclerites, the concentrations of rare elements in the center should match those in the sclerites. However, this study found that the high phosphorus and sulfur concentrations observed in the sclerite-containing coenenchyme on the axis surface were not coincident with their concentrations in the center of the axis. While Tamenori et al. (2014) did not analyze the center of the axis, they reported that concentrations of magnesium and sulfate were high in the sclerites of the P. elatius coenenchyme they analyzed by µ-XRF/XAS mapping. On the other hand, Perrin et al. (2015) reported high magnesium and low sulfur concentrations in sclerites in the central part of the axis in P. elatius, as in C. rubrum. High magnesium and low sulfur concentrations have also been reported in sclerites removed from the coenenchyme of C. rubrum.(Vielzeuf et al., 2013; Vielzeuf et al., 2018). Thus, the present study and Tamenori et al. (2014) differ from the other studies in finding that the sulfur content was high in sclerites in the coenenchyme around the axis surface. To clarify this difference and identify the exact degree of similarity between the sclerites in the coenenchyme and those in the center of the axis, future analysis is required of the trace elements in sclerites removed from both the coenenchyme on the surface and the calcite cement in the center of the axis, whose chemical composition differs from the peripheral part (Vielzeuf et al., 2018). Moreover, regarding the difference between the sclerites and axis of C. rubrum, Le Roy et al. (2021) reported that their protein composition is different. Therefore, it is possible to identify the degree of similarity between the sclerites and the center of the axis by analyzing the protein composition of the center.

Conclusions

This study showed that the growth-ring formation and the concentrations of magnesium, sulfur and phosphorus in the axis of C. japonicum cultured in situ for 98 months were related to the habitat water temperature both qualitatively and quantitatively. The study thus provided basic knowledge to clarify the effects of ocean warming due to global warming on the biomineralization of precious corals. Going forward, it will be necessary to establish experimental techniques for the in-situ and ex-situ culture of precious coral and to proceed with physicochemical and biological research on the selective uptake of each trace element and its relationship with axis growth as well as the influence of environmental factors.

Supplemental Information

Supplemental Information 1 Data of trace elements in the cross section of the Corallium japonicum axis using µ-XRF/XAS mapping analysis for Figure 4.

Click here for additional data file.

Supplemental Information 2 Data of trace elements in the cross section of the Corallium japonicum axis using µ-XRF/XAS mapping analysis for Figures 6-8.

Data is along a transect which is drawn from the center to the surface on a cross section in Figure 5.

Click here for additional data file.

Supplemental Information 3 Data of trace elements in the longitudinal section of the Corallium japonicum axis using µ-XRF/XAS mapping analysis for Figure 9.

Click here for additional data file.

Supplemental Information 4 Data of water temperature on the seafloor at 135 m depth off Takeshima Island, Kagoshima, Japan, from 4 Nov. 2012 to 28 Jan. 2014.

Click here for additional data file.

We would like to thank Dr. Toshio Ninomiya and Dr. Luan Trong Nguyen for their valuable contribution to the study. We also thank Fukada Salvage & Marine Works Co., Ltd. and the crew of their manned submersible Hakuyo for their assistance in operating the in-situ experiment. We are grateful to Dr. Daniel Vielzeuf and an anonymous reviewer for their thoughtful reviews and helpful comments.

Additional Information and Declarations

Competing Interests

Author Contributions

Field Study Permissions

Data Availability

Mutsuro Kikunaga is an employee of the Fukada Salvage & Marine Works Co., Ltd. The other authors declare that they have no competing interests.

Nozomu Iwasaki conceived and designed the experiments, performed the experiments, analyzed the data, prepared figures and/or tables, authored or reviewed drafts of the article, and approved the final draft.

Hiroshi Hasegawa conceived and designed the experiments, performed the experiments, analyzed the data, authored or reviewed drafts of the article, and approved the final draft.

Yusuke Tamenori conceived and designed the experiments, performed the experiments, analyzed the data, prepared figures and/or tables, authored or reviewed drafts of the article, and approved the final draft.

Mutsuro Kikunaga conceived and designed the experiments, performed the experiments, prepared figures and/or tables, and approved the final draft.

Toshihiro Yoshimura performed the experiments, authored or reviewed drafts of the article, and approved the final draft.

Hikaru Sawai performed the experiments, authored or reviewed drafts of the article, and approved the final draft.

The following information was supplied relating to field study approvals (i.e., approving body and any reference numbers):

A permission to collect precious corals was obtained from the Governor of Kagoshima Prefecture, Japan (KAKOSANGO 19-1).

The following information was supplied regarding data availability:

The raw data is available in the Supplemental Files.

The trace element maps in Figures 3 and 4 were drawn based on the Files S1 and S2, respectively. The temperature fluctuation in Figure 5 was drawn based on the File S3.

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
