# Peer review of "Synchrotron µ-XRF mapping analysis of trace elements in in-situ cultured Japanese red coral, Corallium japonicum"

_PeerJ, doi:10.7717/peerj.13931_

## Round 0.1 · original submission · Major Revisions

I have heard back from two expert reviewers, both of whom feel your work has merit, but also have provided many constructive comments that should help you in revising your work. In particular, please consider showing rings via figures more clearly, and some more explanation of some of your methods and decisions. Of course, please consider all comments from the reviewers.

I look forward to seeing a revised version of your work.

Reviewer 1 ·

Basic reporting

Improvement can be made in the construction of some paragraphs, see "Additional comments" for further details.

Experimental design

Experimental design justified but needs some refinement, e.g reason as to why only one colony was used. Also, would be ideal to provide a quantitative test to support the relationship between elemental mapping and seawater temperature. See "Additional comments" for further details.

Validity of the findings

Limited replication but this may be justifiable given the nature of the study, but authors please provide this justification. See "Additional comments" for further details.

Additional comments

Iwasaki et al provide measurements of growth rates of a species of precious coral using in situ specimens, and mapped the distribution of magnesium, phosphorus and sulphur in the skeleton of the coral. The authors then relate the environmental seawater temperature from the habitat where corals were growing with the distribution of skeletal elements and propose relationships. They suggest that Mg, P and S incorporation into the skeletons is related to temperature variability.

This is an interesting study that contributes to the knowledge of the biology of a species of considerable economic and conservation value and confirms previous studies suggesting the relationship between seawater temperature and Mg incorporation. As such, this study merits publication. There are some aspects of the manuscript that require improvement, including strengthening the relationship between seawater temperature and elemental incorporation and providing clear topic and concluding sentences in some paragraphs.

Specific comments:
Line 132-133: The authors mentioned they collected four coral colonies, but only one colony was used for the synchrotron mapping. Why where the other colonies excluded form the study? Explain.
160: In the methods, please mention the use of the Synchrotron.
208: It is hard to see 10-11 rings in the coral cross section, perhaps may be worth including a new figure with arrows pointing to the bands?
214-218: The overlapping images of Mg and S concentrations with seawater temperature vaguely show a relationship, and wonder whether a quantitative approach can be used to provide support to this relationship? Perhaps a regression/ multiple regression analysis?
219-221: Authors may want to improve the expression of the ideas here because when they say that temperature was decreasing, it may seem that they refer to a lower temperature, when in fact in Nov Dec, the temp plot shows higher temperature during that period.
265-267: Mentioned that concentrations of Mg and temp are positively correlated, but it would be ideal to provide a statistical test to support this statement. Mapping images are good but not completely clear.
278: “In previous studies”: cite the studies.
288: “Vielzeuf et al. (2018)”: Not in reference list!
302-303: Need a clear concluding sentence for this paragraph.
304: Improve the topic sentence of this paragraph, particularly since this study does not provide strontium measurements and the reader is wondering why you are discussing this topic.
312-317: The observation of more than one pair of bands (light and dark) during an annual cycle may indicate the relationship with temperature is unclear or weak. Does this challenge the significance of the relationship and consequently of the study?
348: The conclusion may be correct but would be good to include some sort of quantitative relationship between these two variables.

·

Basic reporting

This paper focuses on a colony of precious coral (Corallium japonicum) that was collected, observed, cultured at sea at 135m depth and monitored for a little over eight years. This required large technical means including the use of the submersible Hakuyo. The objective is to observe the growth rings that developed during this period, to characterize them from a chemical point of view, and to see if there are correlations with the temperature of the sea water that was also monitored nearby during part of the experiment.
The objectives of the study and its positioning in current scientific issues are clearly stated. The text is short, in good English, with a correct structure, it reads well. This work comes from a Japanese team recognized in this field and author of outstanding articles on the subject. The figures are not of sufficient quality in terms of definition, but this may be due to the pdf version I had. They are all useful, but not always convincing.
In this work, there is a lack of information on the follow-up and fate of the colony during the experiment. It appears that some branches disappeared during the eight years and one would like to have a series of images illustrating the change in size and shape during the eight years.
For their study, the authors focus their attention on a secondary branch that developed over the eight years. The branch was cut perpendicularly and transversely to prepare sections for study. Light microscopy was used to identify growth rings, and sophisticated techniques such as chemical mapping of major and trace elements by Synchrotron μ-XRF were used.

Experimental design

Regarding the identification of growth rings by optical microscopy, although the technique is effective and recognized, including as a result of the work of the team that signs this study, the images that are proposed to us are not convincing. It is difficult to recognize the growth rings, and the authors do not indicate on their images what they consider to be growth rings. In this kind of study, it is useful to show a large sample area that allows to see the continuity of the growth rings. The conclusions on the radial growth rate suffer from this lack of precision. Moreover, it is difficult to talk about radial growth rate without talking in more detail about the growth rate of the medullary zone which grows in a different way than the annular zone. If this difference is of little importance on old colonies (with a large diameter), it becomes critical on young colonies or branches. This could partly explain the rapid growth rate determined by the authors compared to previous studies (well cited in the article).
As far as chemical characterization is concerned, the technique used does not seem to me to be adapted to the objectives pursued, and using the synchrotron to make this type of chemical imaging is a bit like 'taking a sledgehammer to crack a nut' with the risk of totally crushing the nut. The authors propose chemical images of magnesium, sulfur and phosphorus with a spatial resolution that is rather low (10 to 40 µm). It is surprising that the authors did not use less expensive and more precise techniques such as the electron microprobe which allows both imaging (scanning electron microscopy) and chemical mapping with a much better resolution (the Japanese JEOL electron microprobe has astounding imaging and analytical capabilities). The authors have an interesting and rare sample that would allow them to study in great detail the medullary zone with its sclerites and its very particular cement, or even the growth rings.

Validity of the findings

As far as the conclusions are concerned, the authors speak of chemical correlations between magnesium and phosphorus, and of anticorrelation between magnesium and sulfur. Concerning the first correlation, the authors do not discuss enough the position of phosphorus in C. japonicum. For some authors (Cusack et al 2008; Vielzeuf et al. 2013) phosphorus would be contained in the organic matter of biominerals and would be a good indicator of it. In the Mediterranean red coral (C. rubrum) there is indeed a positive correlation between P and S but only in the organic matter. In the mineral part, the S content is high but the P content is low. The position of P in C. japonicum should therefore be discussed by the authors. Regarding the anticorrelation between Mg and S, it has already been noted in different studies and a literature review would be welcome here. The authors then note that high Mg levels (and low P and S levels) correspond to growth periods when the water is warm and conversely for growth periods when the water is cold. These are also the conclusions of previous studies on different species of Corallium.
Finally, the authors consider that there is a direct relationship between Mg content and temperature and report similar conclusions for Mediterranean red coral. This question is the subject of discussion. While some people favor a direct relationship, others consider that the problem is more complicated. For example, some researchers have investigated whether red corals that grew at different depths and temperatures (monitored temperatures) had different magnesium compositions. The result is that there is no statistically significant difference in composition between these different corals. We regret that the authors of the article do not mention these studies. On the other hand, teams of researchers have also studied the differences in composition observed between the medullary and annular zones of precious corals, knowing that the medullary zones grow fast (about ten times faster than the annular zone). Significant and systematic differences were observed. The conclusion was that the incorporation of magnesium in the skeleton depends on the growth rate of the coral. This type of work may be worth discussing, even contradicted, but should not be ignored by the authors. Complex systems depend on a whole series of parameters and it is not enough to determine this or that type of correlation, it is also important to establish the links between the parameters. Thus for some authors the first order parameter is the growth rate which depends on other parameters such as nutrient abundance, light or temperature. The authors should enrich the reflection on the subject with a more in-depth discussion of the subject.

Additional comments

Some detailed remarks
Line 24: replace 'calcium carbonate' with 'calcite'.
Line 72: determinations of organic matter contents in C. japonicum are available for C. japonicum (Vielzeuf et al 2018).
Line 84: use italics for C. rubrum.
Line 100-101: correlation tests between temperature and chemical composition have been published for red coral (Vielzeuf et al 2013).
Line 157 and 158: we would like more detail on these sclerites. Are they sclerites or microprotuberances? A better picture would be desirable.
Line 211: giving two digits after the decimal point for a temperature when the variation is plus or minus one degree is not significant.
Line 217: This estimate assumes a linear growth during the year. This aspect should be discussed because some authors consider that the growth rates change during the year.
Ligne 250-252: This statement is not true for all elements. It has been verified only for Ba and Pb, and possibly for Li and U. The difference of composition between annular and medullar zones is not considered either.
Line 269: Perrin instead of Perrom?
Lines 279-282: This is not the first study showing elemental maps of entire sections, tranversal or longitudinal.
Line 335-336: The concentration of elements in the medullar zone does not match the one of the sclerites because the ‘cement’ surrounding the sclerites has a particular composition (see Vielzeuf et al 2018) who discuss specifically this aspect.
Line 339-341: this relation between Mg and S in sclerites of P. elatius has been contradicted by Perrin et al. 2015, their Fig. 7. The concentrations of Mg are high and the concentration of S are low.
Line 342-344: it is possible to compare the chemistry of sclerites with medullar cement or annular rings, by looking at the sclerites in the medullar zone, in polished sections (e.g. Perrin et al. 2015, Fig. 7).
References: many references listed in the text are not in the list. References to other important works on the chemistry of precious corals are missing ((e.g. various works on the red coral by Chaabane et al) but difficult to check due to incomplete listing in the reference list).

---

## Round 0.2 · Minor Revisions

I have heard back from one reviewer, who had reviewed your first version, and they find the paper much improved, while offering some small comments. I agree with their assessment, and believe you will be able to easily revise your work.

·

Basic reporting

The authors of this paper on the Japanese red coral took serious care of the comments, and I consider that the paper can be accepted as it is now.
However, I have some minor remarks and additional comments the authors might be interested in.

Line 197-198: ‘a clear negative correlation clear’

Lines 215-221: I think that it is exaggerated to give measurement with a two digit precision, when the error that can be assigned to the measurement is bigger.

Line 319: Please do not quote Vielzeuf et al 2013 for variations of Mg related to fluctuations in temperature. This is not what they are saying.

Line 378: ‘Lauan’

Caption of figure 3 is missing

Experimental design

no comment

Validity of the findings

no further comment

Additional comments

Line 220-221: The studied sample grew during about 8 years (98 months) and 11 growth rings are observed. Knowing the medullar zone takes some time to develop (some authors say 4 years) we would expect a lower number of growth rings. I do not remember seeing a discussion on this (interesting) discrepancy.

Line 348-349: More should be said about the annual variation of temperature of the Pacific waters at these depths. According to the data that are presented, the variation of temperature during the year seems pretty small. Furthermore the difference of temperature for the two selected periods (Nov-Feb and Feb-May) is small. Thus conclude that the origin of the variation of composition in the growth rings is due to temperature variations is pretty thin. Growth rings exist in the Mediterranean red coral in colonies collected at great depths where there is no variation of seawater temperature. I understand that common belief in our community is that variations in Mg contents are temperature dependent but these data seem to indicate otherwise. Note that I am not asking the authors to change their mind on the conclusion, it is just a personal observation.

---

## Round 0.3 · accepted · Accept

Thank you for your revision, and I am pleased to move this work into production.